# Transcriptional Coactivator BOB1 (OBF1, OCA-B) Modulates the Specificity of DNA Recognition by the POU-Domain Factors OCT1 and OCT2 in a Monomeric Configuration

**DOI:** 10.3390/biom14010123

**Published:** 2024-01-17

**Authors:** Igor B. Nazarov, Danil S. Zilov, Mikhail N. Gordeev, Evgenii V. Potapenko, Nataliya Yeremenko, Alexey N. Tomilin

**Affiliations:** 1Institute of Cytology, Russian Academy of Sciences, Tikhoretsky Ave. 4, 194064 St. Petersburg, Russia; zilov.d@gmail.com (D.S.Z.); misha.n.gord@gmail.com (M.N.G.); 2Institute of Evolution, University of Haifa, Haifa 3498838, Israel; potapgene@gmail.com; 3University of Haifa, Haifa 3498838, Israel; 4Center for Research in Transplantation and Translational Immunology UMR1064, 30 Bd Jean Monnet, Nantes University, CEDEX 01, 44093 Nantes, France; nataliya.yeremenko@univ-nantes.fr

**Keywords:** BOB1, OCA-B, OBF1, OCT1, OCT2, transcription factor specificity, autoimmune diseases, hematologic malignancies, germinal center (GC)-derived lymphomas

## Abstract

BOB1, a mammalian lymphocyte-specific transcriptional coactivator of the transcription factors OCT1 and OCT2 (OCT1/2), plays important roles in normal immune responses, autoimmunity, and hematologic malignancies. The issue of a DNA sequence preference change imposed by BOB1 was raised more than two decades ago but remains unresolved. In this paper, using the EMSA–SELEX–Seq approach, we have reassessed the intrinsic ability of BOB1 to modulate the specificity of DNA recognition by OCT1 and OCT2. Our results have reaffirmed previous conclusions regarding BOB1 selectivity towards the dimer configuration of OCT1/2. However, they suggest that the monomeric configuration of these factors, assembled on the classical octamer ATGCAAAT and related motifs, are the primary targets of BOB1. Our data further specify the DNA sequence preference imposed by BOB1 and predict the probability of ternary complex formation. These results provide an additional insight into the action of BOB1—an essential immune regulator and a promising molecular target for the treatment of autoimmune diseases and hematologic malignancies.

## 1. Introduction

BOB1 (OCA-B, OBF1) is a lymphocyte-specific transcriptional coactivator of the POU-family class II transcription factors (TFs) OCT1 and OCT2 (OCT1/2, POU2f1/2), playing an important role in the regulation of immune responses in both physiological and pathological contexts. An aberrant expression of BOB1 is associated with multiple autoimmune and chronic inflammatory diseases (reviewed by Yeremenko et al. [1]) and germinal center (GC)-derived lymphomas [2]. The BOB1 and OCT1/2 factors are extensively colocalized and bind widely to the promoters or enhancers of genes involved in the germinal center (GC) formation in mice and humans, orchestrating the transcriptional program of this secondary lymphoid organ [3,4]. However, the detailed mechanisms of their cooperative action remain largely unclear. 

There is widespread degeneracy in DNA sequence recognition by eukaryotic TFs [5,6,7], including those of the POU family [8]. Moreover, the modulation of sequence recognition during the interaction of two or more TFs with DNA can be one of the most crucial combinatorial mechanisms of transcriptional regulation. It has been demonstrated that the majority of TF pair sites involve a large overlap between the recognition motifs of individual TFs, resulting in the recognition of composite sites that markedly differ from the individual TF motifs [9]. This principle fully applies to the cooperation between OCT1/2 and BOB1, which was shown more than two decades ago [10,11,12,13,14] but has not been explored in detail since that finding. It has been demonstrated that while OCT1/2 binds the well-known canonical octamer ATGCAAAT and related sequences, only a subset of these sequences allows the formation of ternary complexes with BOB1. Of 10 octamer-derived OCT1/2-binding sites, only four are permissive for BOB1. In addition, a stringent requirement for adenine in the fifth position of the octamer motif has been demonstrated for the recruitment of BOB1 by OCT1 [11]. The other classes of POU domain-interacting DNA elements, such as the recently discovered CpGpal [12], have to be yet examined for their ability to recruit BOB1. A notable attempt to explore the specificity of the ternary complex formation was made 20 years ago using the derivatives of the palindromic Oct factor recognition element (PORE) sequence ATTTGAAATGCAAAT from an intronic enhancer of the *osteopontin* (*Spp1*) gene [14]. The PORE, composed of two inverted octamer motifs overlapped by one nucleotide, interacted with POU-domain factor dimers, thereby mediating strong transcriptional activation [15]. Lins and coworkers generated several oligonucleotides with single-nucleotide mutations in the PORE sequence and demonstrated that the obstructing effects of several mutations that reduce or even abolish the POU complex formation in both monomeric and dimeric configurations were attenuated in the presence of the BOB1 peptide in the ternary complex, which still included two POU molecules [14]. However, the recent studies of BOB1/OCT1/2 complex distribution in mouse [3] and human genomes [4] have found no evidence that PORE-type or novel-class genomic elements are engaged by this ternary complex—the octamer and closely related motifs remained the most preferred binding targets.

In this study, we have reassessed the intrinsic ability of BOB1 to modulate the specificity of DNA recognition by the POU domains of OCT1 and OCT2 using an in vitro approach to random selection. Our results have shed additional light on the mechanism of BOB1 action in mammals.

## 2. Materials and Methods

### 2.1. Plasmid Constructs, Expression, and Purification of Proteins

The schematic representations of the plasmid constructs used for the expression of double-tagged POU1, POU2, and BOB1 proteins in *E. coli* are shown in Appendix A. Plasmid constructs were verified by sequencing to confirm the absence of errors. Protein expression was induced with 1 mM IPTG in *E. coli* BL21 cells (BL21[DE3]pLysS, Novagen; Madison, WI, USA) for 12–18 h at 25 °C. The proteins were purified under denaturing conditions and then renatured using long, stepwise dialysis. For this procedure, bacterial cells were lysed in buffer A1 (8 M urea, 0.1 M sodium phosphate buffer, 0.01 M Tris–HCl pH 8.0, 5 mM β-mercaptoethanol). The clarified cell lysate was incubated with Ni–NTA His-Bind Resin (Novagen) for 2 h at RT, washed with buffer A2 (8 M urea, 0.1 M sodium phosphate buffer, 0.01 M Tris–HCl pH 6.3, 5 mM β-mercaptoethanol), and eluted with buffer A3 (8 M urea, 0.1 M sodium phosphate buffer, 0.01 M Tris–HCl pH 4.5, 5 mM β-mercaptoethanol). The protein samples were then subjected to four-stage dialysis at 4 °C against buffers with a two-fold decrease in the concentration of urea and a gradual increase in pH at each stage. The final dialysis step was performed against PBS containing 5 mM β-mercaptoethanol for 12 h. To obtain homogeneous proteins essentially free of truncated forms and contaminating proteins, the POU1 and POU2 proteins were further purified using Pierce Streptavidin Plus UltraLink Resin (Thermo Fisher Scientific; Waltham, MA, USA), whereas BOB1 protein was additionally purified using ANTI-FLAG M2 Affinity Gel (Merck, Rowe, NJ, USA). The purification conditions were set according to the manufacturer’s recommendations. POU1 and POU2 were eluted in a buffer containing 10 mM biotin. BOB1 was eluted in a low-pH buffer. The resulting proteins were analyzed using SDS–PAGE (Appendix A), and their DNA-binding activity was verified using electrophoretic mobility shift assay (EMSA).

### 2.2. EMSA–SELEX

Oligonucleotide 5′–ACGACGCTCTTCCGATCT–N_35_–GATCGGAAGAGCACACGTC–3′, with a total length of 72 nucleotides, was synthesized by Evrogen (Moscow, Russia). Single-stranded DNA was converted to a double-stranded form and amplified with PCR using specific primers and Phusion High-Fidelity DNA Polymerase (Thermo Scientific). The binding reaction was performed by mixing 300 ng (6.42 pmol) of SELEX-DNA with 30 ng of POU1 (0.93 pmol) or POU2 (0.83 pmol) protein and, when specified, 28 ng (0.8 pmol) of BOB1 protein in 20 μL of binding buffer (20 mM Tris–HCl pH 7.6, 10% glycerol, 0.14 M NaCl, 0.5 mM EDTA, 0.05% Triton X-100, 0.1 mg/mL BSA, 0.1 mg/mL poly[dI–dC] (Amersham, UK), and 5 mM DTT), then incubated for 20 min at RT. Next, the samples were separated using native 6% PAGE in 0.5× TBE buffer at RT. Gel fragments were excised from areas corresponding to complexes whose mobilities had been visualized in a parallel EMSA with SELEX-DNA replaced by a fluorescein-labeled PORE^D^ oligonucleotide [15,16]. After the elution of the SELEX-DNA from the gel, fragments were amplified using PCR, precipitated with ethanol, and used for subsequent enrichment cycles. After four rounds of EMSA–SELEX, the samples were sequenced using an Illumina Solexa Hiseq1500 system at Genoanalytica (Moscow, Russia). Motif elicitation was performed with the STREME online tool [17].

## 3. Results

Sequences encoding the tagged DNA-binding POU domains of human OCT1 (POU1) and OCT2 (POU2), as well as the full-length human BOB1 protein, were cloned into the pET28a vector (Appendix A) and expressed in *E. coli* BL21 cells. The recombinant proteins were purified to homogeneity (Appendix A) and verified by electrophoretic mobility shift assay (EMSA) for the ability to form ternary complexes with DNA. Subsequently, the proteins were used for the enrichment of specific DNA sequences from complexes with POU1, POU1/BOB1, POU2, and POU2/BOB1, using the EMSA-based systematic evolution of ligands by exponential enrichment followed by sequencing (EMSA–SELEX–Seq) procedure. The approach included consecutive rounds of the enrichment of DNA sequences, mediating the formation of ternary complexes with BOB1 and POU1/POU2. To this end, DNA oligonucleotides containing random 35-nucleotide sequences flanked by defined sequences (designated for both amplification and NGS sample preparation) were synthesized. The complexes that formed were separated using EMSA (Figure 1). After four rounds of enrichment and amplification via the EMSA–SELEX procedure, the samples were sequenced.

After filtering and trimming the raw DNA reads, we obtained four sets of 35-nucleotide sequences, with each set ranging from approximately 140,000 to 275,000 (Appendix A). To elicit all possible motifs, the STREME online tool [17] was applied to these sets. Several motifs, varying in frequency and *p*-value, were revealed (Figure 2). As expected, the leading motif represented the well-known octamer site A_1_T_2_G_3_C_4_A_5_A_6_A_7_T_8_; however, the tool generated this site as two separate output motifs (Figure 2a vs. Figure 2b). Nevertheless, both of their frequencies increased upon the addition of BOB1, from 48.7% to 70.0% and from 8.4% to 13.0% for complexes with POU1, as well as from 39.7% to 61.4% and from 6.0% to 10.8% for complexes with POU2 (Figure 2a,b), consistent with the originally postulated mode of action of this coactivator as a molecular clamp stabilizing the POU–DNA interaction [18]. Another notable change in sequence preference imposed by the presence of BOB1 was the increased frequency of A_5_ from 82% to 98% (POU1) and from 81% to 99% (POU2), which also applied to A_6_, whose frequency increased from 95% (POU1) or 93% (POU2) to 99% in both cases. These data are fully consistent with previous observations, which have shown that A_5_ and A_6_ of the octamer sequence are strongly preferred because they form a pair of hydrogen bonds, while T’s in the complementary DNA strand make hydrophobic contacts with BOB1 [19,20,21]. In addition, the presence of BOB1 in the complexes reduced the probability of T_2_ (and, additionally, of T_8_ for POU1) while increasing the frequency of C_4_ (Figure 2a,b). Overall, POU1 and POU2, either alone or in complexes with BOB1, showed remarkable similarity in the recognition of the octamer site variants as well as of some non-octamer motifs (see below). Our results are consistent with patterns of octamer motifs obtained by different methods employed by other research groups, for example, the motif logos obtained by the consecutive affinity purification and systematic evolution of ligands using exponential enrichment (CAP–SELEX–seq) for POU1 and POU2 without BOB1 [22] as well as chromatin immune precipitation with massively parallel DNA sequencing (ChIP-seq) analysis for mouse OCT1 and human OCT2, either alone or in a complex with BOB1 [3,4]. These results indicate high conservation and consistency in the interactions of these two proteins with BOB1 and DNA as well as the impact of the BOB1 coactivator on sequence preference.

In addition to the classical octamer and related motifs, the STREME online tool revealed less-frequent motifs that resembled the MORE (more of PORE, consensus sequence ATG[C/A]AT[A/T]_0–2_AT[T/G]CAT), which have been previously shown to mediate the formation of POU1 dimers in a refractory to BOB1 configuration [8,16,23]. Fully in agreement with the above studies, the MORE, MORE^+2^, and MORE^+1^ sequences were revealed exclusively in POU1 and POU2 complexes lacking BOB1 (Figure 2c–e). This result, along with the nearly undetectable T_5_ in POU/BOB1 complexes of the octamer site (Figure 2a,b), suggests an efficient separation between BOB1-free and BOB1-containing complexes (Figure 1). It further asserts the overall applicability of the EMSA–SELEX–seq method.

The PORE motif (consensus sequence ATTTGAAATGCAAAT), even though it was revealed using EMSA–SELEX–seq at low frequencies (0.1–0.5%) and, in complex with POU2, in conjunction with *p*-values greater than 0.05 (Figure 2f), deserves special consideration. This motif, previously discovered in an enhancer of the *osteopontin* (*Spp1*) gene, has been shown to bind dimers of POU proteins in a configuration different from that on the MORE motif and to mediate the transcriptional activation by these factors [15]. The PORE-type configuration of the OCT1 dimer, contrary to the MORE configuration, is capable of recruiting BOB1 [16,23]. Our data further substantiate this conclusion, as PORE was represented across all four complexes (Figure 2f). However, in the presence of BOB1, neither an increase in frequency nor a significant modulation in the DNA-binding specificities of the POU factors occurred within both PORE (Figure 2f) and other non-octamer motif contexts (Appendix A). Therefore, our data provide no evidence for the previously stated role of BOB1 in the alleviation of DNA sequence requirements by the POU factors [14].

Next, we sought to inspect the individual octamer-type motifs within the complexes of POU1 and POU2 with or without BOB1 (Figure 2a,b). To this end, we extracted all possible sequence variants matching the pattern ANNNNAAN, determined the frequencies of individual sequences, and analyzed their enrichment in each sequenced set. The chosen pattern accounts for highly conserved A_1_, A_6_, and A_7_ as well as for fairly variable positions 2–5 and 8 within the octamer motif (Figure 2a,b), while minimizing nucleotide-shifted sequences and avoiding reverse complements. Because the total number of DNA reads in the sequencing data sets were different, we calculated the relative frequencies of the individual sequences as percentages. Next, we sorted these sequences in a descending order by different columns (Appendix A). As expected, the top positions in all tables were occupied by the canonical octamer ATGCAAAT (outlined with a rectangle). Canonical octamers represented approximately 5% of POU1 and POU2 complexes, approximately 8% of POU1 + BOB1 complexes, and approximately 13% of POU2 + BOB1 complexes. It should be noted that the presence of BOB1 in the complexes increased the canonical octamer frequency by 1.5–2 times. The top 100 sequences (out of 1024), which constituted a total of approximately 50% of the top frequencies, were selected for further analysis.

The distribution of individual DNA sequences found in POU1 and POU1 + BOB1 complexes was nearly indistinguishable from that in POU2 and POU2 + BOB1 complexes, respectively (Figure 3a,b and Appendix A). In contrast, BOB1 introduced significant perturbations in frequency distribution (Figure 3c,d and Appendix A). As expected, sequences containing T_5_ were significantly underrepresented by both the POU1 + BOB1 and POU2 + BOB1 complexes (Figure 3c,d and Appendix A), consistent with the previous data [19] and the STREME analysis results (Figure 2a,b). In addition, POU1 and POU2 complexes tolerated substitutions in octamer positions 3–5 and to T’s, whereas the corresponding complexes with BOB1 demonstrated a tolerance to substitutions mostly in positions 2 and 8, while T’s were replaced by other nucleotides. In addition, the presence of BOB1 in the complexes led to a 4- and 2.9-fold reduction in the frequencies of the sequence ATTTAAAT as compared to complexes with POU1 and POU2 alone, respectively (Figure 3c,d and Appendix A).

To further correlate motif frequency with the ability to mediate the assembly of DNA/POU/BOB1 ternary complexes in vitro, we performed an EMSA with three sequences selected from Appendix A. These sequences were placed in the context of the immunoglobulin kappa light chain promoter (*Vk*), in which the consensus octamer site occurs naturally. As for the three selected motifs, a direct correlation between the relative frequency and the intensity of the ternary DNA/POU/BOB1 complexes was observed (Figure 4). Overall, the reported sequence frequencies identified by the EMSA–SELEX–seq (Appendix A) seem to be predictive of the affinity of this complex.

## 4. Discussion

BOB1 is an important regulator of the immune response, and its abnormal activity is strongly associated with multiple autoimmune pathologies and GC-derived lymphomas, thus making this protein an attractive target for their treatment [1]. Several attempts have been made to dissect the mechanisms of BOB1′s action as a coactivator of the POU-domain transcription factors—the ubiquitous OCT1 and B cell-specific OCT2. 

In this study, using the EMSA–SELEX–seq approach, we further extended our knowledge of how BOB1 modulates the specificity of DNA recognition by these POU factors. With regard to octamer-related sequences, our data substantiate the view that BOB1 stabilizes the binding to the canonical octamer and related motifs, consistent with a mode of action as a molecular clamp stabilizing POU–DNA interactions [18]. In addition, we have confirmed the stringent requirement for adenine in position 5 [11] and also have shown that thymines in positions 3 and 4 of the octamer motif are permissive for POU1/2, but unfavorable for the formation of a ternary complex with the BOB1 coactivator. Moreover, BOB1 recruitment is favored by substitutions of thymines to one of the remaining three nucleotides, mainly in positions 2 and 8. Many other substitutions with changing or maintained frequencies have also been observed (Appendix A). Thus, it can be concluded that there are many sequences different from the canonical octamer, containing one or more nucleotide substitutions, capable of forming complexes with POU1/2 and that complexes with and without the BOB1 coactivator have different preferences for the DNA sequence of the octamer. Overall, the reported octamer-like sequence frequencies identified by the EMSA–SELEX–seq seem to be predictive of the probability of formation of this complex in vitro and possibly in vivo.

An important part of our study concerns BOB1 recruitment by OCT1/2 dimers. We have reaffirmed the paradigm that the MORE-type OCT1/2 dimer assembly is incompatible with BOB1, while a PORE-type configuration is permissive to this coactivator [16]. However, our EMSA–SELEX–seq results suggest that PORE is not a preferred dimeric configuration and, notably, that BOB1 does not enforce it, as it does not increase its frequency (Figure 2f). In addition, BOB1 does not enforce the emergence of any novel motifs (Appendix A). These observations contrast with the previous finding that BOB1 can markedly alleviate the DNA sequence requirements of the OCT1 dimer in the PORE configuration and that the dimeric rather than the monomeric configuration is the primary target of BOB1 [14]. This apparent discrepancy warrants further investigation; however, the recent studies of BOB1 and OCT1(OCT2) protein distribution in mouse [3] and human genomes [4] have found no evidence of PORE-type or novel class genomic elements being engaged by BOB1/OCT1/2, leading to the conclusion that the octamer and closely related motifs are the primary binding targets of OCT1/2 complexes containing BOB1.

The EMSA–SELEX–seq developed in this study seems to be a highly reliable method to study the DNA binding specificity of transcription factors. First, there are almost indistinguishable patterns of DNA motifs as well as of relative frequencies of individual sequences between complexes containing POU1 and POU2, owing to the high degree of amino acid sequence similarity. Also, nearly identical motif patterns and octamer sequence variants were observed within ternary complexes containing the POU-proteins and BOB1—the inclusion of BOB1 in ternary complexes resulted in almost uniform changes in DNA sequence preference for both POU1 and POU2 (Figure 2 and Figure 3). Second, fully in agreement with the previously established refractoriness of MORE-type POU dimer configuration to BOB1 [8,16,23], the MORE, MORE^+2^, and MORE^+1^ sequences were revealed exclusively in complexes lacking BOB1 (Figure 2c–e). We identified these sequences as MORE variants even though they did not perfectly match the consensus MORE, ATG[C/A]AT[A/T]_0–2_AT[T/G]CAT, which was deduced based on in vivo data [24,25,26]. Supporting our conclusion, an alignment of these motifs with published databases using the Tomtom program (MEME suite 5.5.5) revealed the best similarities to sequences in the database identified as bound by different POU-domain factors [22], albeit without references to their stoichiometry or configuration. We consider two ATs, conjoint or separated by up to two As or Ts with the consensus (AT[A/T]_0–2_AT), to be a hallmark of the MORE—this pattern does not match any other known POU protein-binding motifs. Notably, this study seems to represent the first in vitro attempt to portray sequence preferences by POU dimers specifically in the MORE configuration. Perhaps the most convincing evidence of the high reliability of the EMSA–SELEX–seq method is the strong preference of A over T in the fifth position in the ternary complexes with BOB1 (Figure 2 and Figure 3)—the refractoriness of T_5_ within the octamer site to BOB1 has been repeatedly documented [19,20,21]. Furthermore, because the developed method discriminates protein–DNA complexes with different electrophoretic mobilities, it provides an important advantage over bulk methods such as CAP-SELEX [9,22]. As illustrated in this study, the EMSA–SELEX–seq method allows the selective study of DNA sequence preferences of a transcription factor bound in monomeric or homo-dimeric configuration, as well as in complexes with co-activator.

Correlating the octamer site variants described in this paper with diseased conditions in human, like autoimmune diseases or germinal center (GC)-derived lymphomas, represents a highly relevant pursuit. In our attempt to address this point, we first performed some literature mining. Single nucleotide polymorphisms (SNPs) within OCT1-binding sites that are associated with multiple autoimmune diseases, like celiac disease (CD), human systemic lupus erythematosus (SLE), rheumatoid arthritis (RA), type 1 diabetes (T1D), and primary biliary cirrhosis (PBC), have been reported by Maurano et al. [27]. A SNP (rs1046295) associated with asthma was revealed within the OCT1-binding site of *PHF1* gene [28]. Polymorphisms affecting OCT1 binding to the *TNFα* promoter [29,30], *Il-13* gene enhancer HS4 [31], and *HLA-C* gene enhancer [32] have been described. The latter is particularly intriguing as the high levels of HLA-C are associated with an increased risk of developing Crohn’s disease (CRD) [33]. Similarly, the SNP C/T within the OCT1-binding site located in the promoter of *TNF* gene appears to alter constitutive TNF expression and is associated with the susceptibility to inflammatory bowel disease [34]. The authors have proposed that OCT1 physically interacts with the proinflammatory nuclear factor kappa-B (NF-κB) at an adjacent binding site, inhibiting its transactivating effects and affecting the susceptibility to IBD [34]. Moreover, a study based on the combination of high-density genotyping and epigenomic data revealed that CD-associated SNPs are enriched in OCT2-binding sites [35]. However, the listed data provide little clue about mechanisms by which the functional variants in regulatory elements affect the transcription of disease-associated genes with regard to BOB1 recruitment to the OCT1(2)–DNA complexes. This question could be addressed, for example, by correlating the ChIP-seq data on OCT2 and BOB1 genome binding in B cells [3] and ClinVar NCB database with subsequent referencing to our EMSA-SELEX-seq results.

Our working hypothesis is that an elevated BOB1 expression is involved in the pathogenesis of autoimmune diseases [1] and germinal center (GC)-derived lymphomas [2] through the OCT1(2)-mediated engagement of auxiliary genomic targets. ChIP-seq experiments with human B- and T-cells overexpressing BOB1 or having the BOB1/OCT1(2) axis disturbed with the help of our proprietary small molecule inhibitor (patent application: WO2023180387A1) are currently underway. This knowledge should provide an important insight into the mechanism of action of BOB1—a key immune regulator and a promising molecular target for the treatment of autoimmune diseases and hematologic malignancies.

## Figures and Tables

**Figure 1 biomolecules-14-00123-f001:**
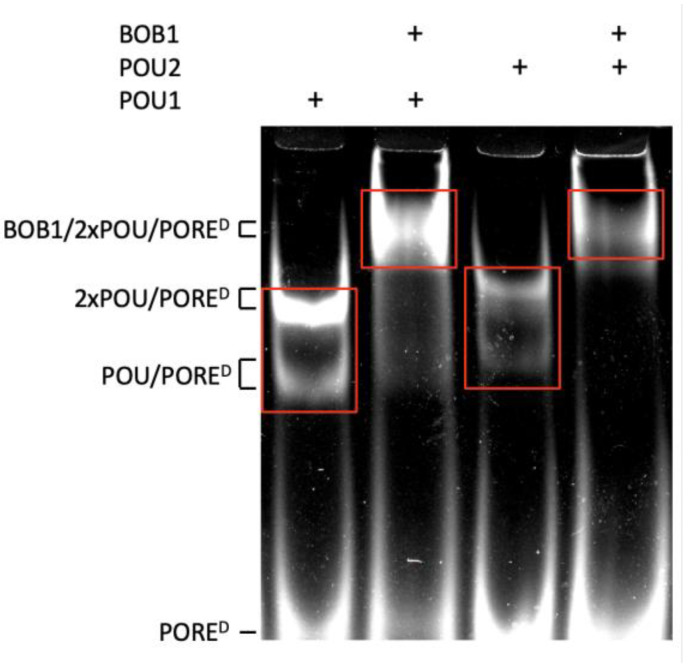
Separation of POU1, POU1/BOB1, POU2, and POU2/BOB1 complexes using EMSA. A fluorescein-labeled double-stranded oligonucleotide containing PORE^D^ [15,16] was used as a reference for complex mobilities. In a parallel EMSA, which was run with the same recombinant proteins but random oligonucleotides, the zones marked with red rectangles were excised and used for DNA extraction and amplification during four consecutive rounds of the EMSA–SELEX procedure, ending with NGS sequencing. Original image can be found in Appendix A.

**Figure 2 biomolecules-14-00123-f002:**
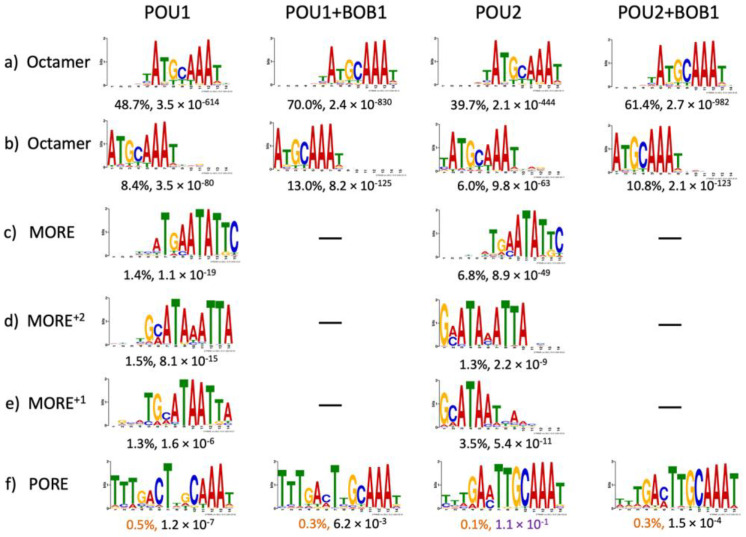
DNA sequence motifs found to form complexes with the POU domains of OCT1 (POU1) or OCT2 (POU2) and BOB1 using the EMSA–SELEX–seq approach and the STREME online tool (as indicated above) [17]. Indicated under each logo are motif frequencies (%) and *p*-values for the complexes specified above the columns. Motifs listed in this figure exhibited frequencies above 1% and *p*-values below 0.05 (**a**–**e**), except for the POREs (**f**) which exhibited frequencies below 1% (colored with orange) and, for POU2, *p*-value above 0.05 (colored with purple). Motifs that cannot be readily classified as POU-domain binding elements are listed in Appendix A.

**Figure 3 biomolecules-14-00123-f003:**
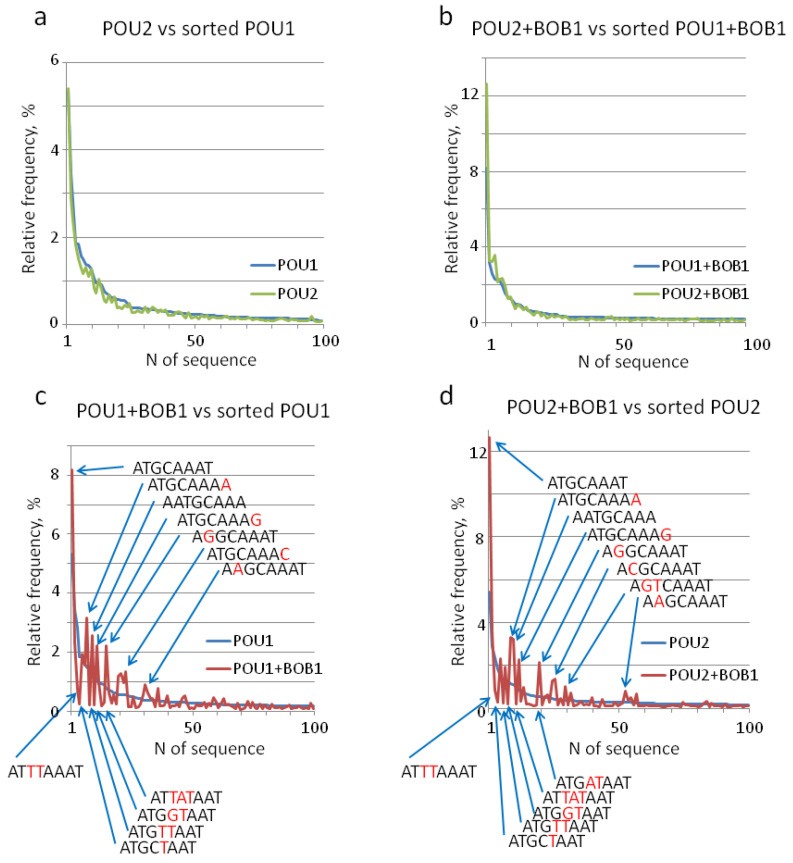
Comparative analysis of relative frequencies of individual sequences enriched by the EMSA–SELEX–seq in complexes with POU1, POU2, and BOB1. The top 100 sequences matching the ANNNNAAN pattern were sorted in a descending order by relative frequencies and compared pairwise on the plot as indicated above each chart. The relative frequencies of individual octamer sequence variants, while comparing DNA samples from complexes with POU1 and POU2 (**a**) and from complexes with POU1 + BOB1 and POU2 + BOB1 (**b**), were nearly indistinguishable. On the contrary, a similar comparison of DNA sequences from POU1+BOB1 and POU1 (**c**) and from POU2 + BOB1 and POU1 complexes (**d**) produced positive and negative peaks, indicating a significant influence of BOB1 presence on the relative frequencies of several sequences. Some of these peaks are denoted, while the complete list of individual sequences with their relative frequencies in these pairs can be found in Appendix A. Nucleotides different from those in the canonical octamer motif (ATGCAAAT) are colored in red.

**Figure 4 biomolecules-14-00123-f004:**
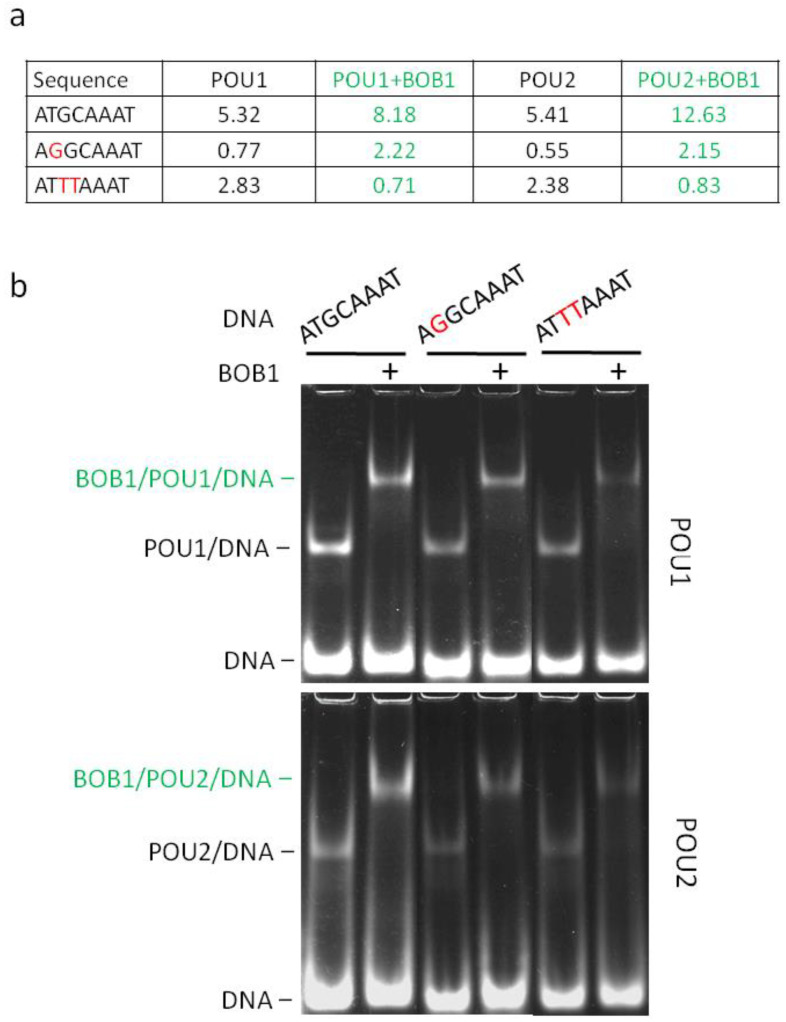
Direct correlation of the sequence frequencies established by EMSA–SELEX–seq and the ability to mediate the formation of ternary DNA/POU/BOB1 complexes in vitro. (**a**) Table reporting relative frequencies (%) across the four sequence groups of the consensus octamer site and its two mutants (deviations are in red). (**b**) An EMSA showing the formation of ternary complexes (in green) and comprising POU1 (upper panel) or POU2 (lower panel), BOB1, and a fluorescein-labeled oligonucleotide containing one of the indicated sequences. The EMSA conditions were the same as those described in Section 2.2 (EMSA–SELEX) except that the amount of DNA was 200 ng (8 pmol), making the DNA/protein molar ratio about 10:1. The signals were quantified using densitometry. Pearson correlation coefficients between the relative frequencies of the individual sequences and the values of the signals from the complexes were calculated. For POU1 and POU2 complexes, the coefficients were determined as 0.863 and 0.948, respectively. For POU1/BOB1 and POU2/BOB1 complexes, they were 0.938 and 0.998, respectively. Each EMSA was performed in two replicates. Original images can be found in Appendix A.

## Data Availability

All raw data of this manuscript are presented in Appendix A.

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
