# Peer review of "Transcriptional Coactivator BOB1 (OBF1, OCA-B) Modulates the Specificity of DNA Recognition by the POU-Domain Factors OCT1 and OCT2 in a Monomeric Configuration"

_biomolecules, 2024, doi:10.3390/biom14010123_

Round 1
Reviewer 1 Report
Comments and Suggestions for Authors
POU factors have bi-partite DNA binding domains and can target a diverse array of composite binding sites. PTMs and co-factor can influence affinity and latent specificity. Authors build upon previous findings and re-valuate how the non DNA binding co-factor influences DNA recognition of OCT1 and OCT2 using EMSA-SELEX. They make the interesting finding, that BOB1 apparently increases the binding affinity of OCT1/2 to octamer DNA – with subtly changing specificity. I find the study interesting but figures and text are hard to read and should be substantially improved. I also recommend adding more quantitative binding data.
Major points:
1. The findings would be stronger if the authors could quantify the effect of BOB1 on OCT1/2 binding. For example, binding isotherms in the absence/presence of BOB1 or related means of quantification. Some optimization of their existing EMSA method should allow this. Solely relying on SELEX data to make the claim for an affinity enhancement is insufficient.
2. Similarly, the effect of the change of A5 should be studied using more quantitative methods. I’m actually quite confused about this point: ‘other remarkable sequence preference change imposed by the presence of BOB1 was the decreased frequency of A5 from 82 to 98% (POU1) and from 81 to 99% (POU2), which also applied to A6 whose frequency increased from 95 (POU1) or 93 (POU2) to 99% in both cases.”
Minor points:
1. Authors may consider the Spec-seq method developed by Gary Stormo in future studies which allows for a more quantitative dissection of binding.
2. To quantify dimer formation. Authors could estimate homodimer cooperativities (omega values) at a steady state.
3. Please clarify in all figures how often experiments were replicated (n = ..) and if appropriate, quantify gels using densitometry (i.e. Figure 1).
4. I thought the MORE motifs (ATGCATATGCAT) in Figure 2 are rather cryptic. Please comment. Authors could perform motif scanning for a targeted search for known moitfs.
5. The authors have added fundamental insights into the DNA recognition of POU factors which determines their selectivity. Aside from octamer, PORE, MORE, NORE POU some POU factors can also form a dimer on a methylated CpGpal site which could be introduced and discussed.
6. EMSA-SELEX is very interesting but the analysis and representation of data could be improved. The authors could seek inspiration from studies of the Jussi Taipale lab (HT-SELEX).
7. Writing and figures can be improved for clarity and better alignment.
8. What are the differences between the octamer motifs in 2a and 2b? The below is not aligned with the figure: “Remarkably, the motif increased in frequency in complexes containing BOB1 from 57 to 83% 144 (POU1) or from 46 to 71% (POU2)” adding up percentages of near-identical motifs is not a valid approach.
9. The analysis in figure 3 is interesting but the representation is confusing and tough to digest. I suggest a scatter plot of normalized counts for given sequences (i.e. OCT vs OCT/BOB). Packages such as DESeq2 could be considered 0 it doesn’t work just for RNAseq but also for oligo count data. I don’t understand Figure 3a/b. Figure 3c/d appears to contain the interesting information.
10. Figures and figure legends are not self-explanatory and should be improved.
11. Figure 3b why BOB1 vs Bob1? Are human/mouse proteins compared?
12. Figure 4: I am not convinced about the conclusions. Please clarify replication. Please show whole gels – do not crop free DNA. (a) should be presented as bound fraction. I recommend titrations and Kd estimates in the absence and presence of BOB. Suggest to use a low DNA concentration. Recommended reference:https://elifesciences.org/articles/57264
13. Figure 4a needs statistics. If authors do not perform the proposed KD titrations they could quantify the fraction bound over the whole lane and present data a log2FC(OCT-BOB/BOB) with n>3 repeats and stats test.
14. Provide more details (i.e. molar concentrations) for all EMSAs.
Comments on the Quality of English Languageextensive editing recommended
Author Response
Please, see attachment

Reviewer 2 Report
Comments and Suggestions for Authors
I have attached my review comments as word document.

Author Response
Please, see attachment

Round 2
Reviewer 1 Report
Comments and Suggestions for Authors
The authors made some improvements and added clarifications to the manuscript. Whilst, unfortunately, not all of my initial comments were addressed with more analyses and experiments because of the overall interest as to how sequence features affect co-factor recruitments by POU factors I am in favour of publication.
Author Response
We thank the Reviewer for spending his/her time and effort for the improvement of our manuscript.
Reviewer 2 Report
Comments and Suggestions for Authors
Manuscript ID: biomolecules-2744228
In the revised manuscript entitled “Transcriptional coactivator BOB1 (OBF1, OCA-B) modulates the specificity of DNA recognition by the POU-domain factors OCT1 and OCT2 in a monomeric configuration” the authors have replied to most of my questions. Though they have not commented on couple of major clarifications like extending the in vitro analysis by bioinformatics tools to report the presence of any mutant motifs in disease conditions or discussing if any of these mutations are hot-spot mutations in the promoters and enhancers of BOB1 regulated genes. As I said earlier, discussing these details may clarify the biological significance of the study, and would benefit far-ranging readers in this field. In Figure S1, the authors have added the molecular weight as requested, but the sentence is repeated twice.
Overall, the manuscript can be considered for publication with the clarifications requested.
Author Response
First of all, we apologies for unintentionally skipping the major point about correlation of our data with mutant motifs in diseased conditions raised by Reviewer 2 in Review Report (Round 1). Somehow it slipped out of our response. This is, of course, a valid point and highly relevant pursuit, which could be partly addressed, for example, by correlating the ChIP-seq data on OCT2 and BOB1 binding in B-cell genome [3] and ClinVar NCB database with subsequent referencing to our EMSA-SELEX-seq results. Performing this task, unfortunately, is beyond our current capacity for multiple reasons, including the very limited period allowed for this revision and lack of available bioinformaticians during this period of year.
In our limited attempts to somehow address the point raised by Reviewer 2 we performed some literature mining. SNPs within OCT1 binding sites, which are associated with multiple autoimmune diseases have been reported by Maurano et al., 2020 (doi: 10.1126/science.1222794). Yet, there is no specific information about the motifs where these SNPs occur neither in the body of the paper nor in the Supplemental data. A SNP (rs1046295) was revealed within OCT1-binding site of PHF1 gene associated with asthma (Holt et al., 2011, doi: 10.1016/j.jaci.2010.12.015). Polymorphisms affecting OCT-1 binding to the TNFalpha promoter (Knight et al. 1999, doi: 10.1038/9649; Hohjoh et al., 2001; 10.1038/sj.gene.6363721), Il-13 gene enhancer HS4 (rs1881457, Kiesler et al., 2009, doi: 10.1093/hmg/ddp411), and HLA-C gene enhancer (rs2395471, Vince et al., 2016; doi: 10.1016/j.ajhg.2016.09.023). These limited literature data, however, provide little information about how the functional variants in a regulatory element affect transcription of disease-associated gene with regard to BOB1 recruitment to the OCT1(2)-DNA complexes.
Anyway, we thank the Reviewer for bringing our attention to this very important issue, which will be addressed in our future studies. Our current hypothesis is that elevated BOB1 expression is involved in pathogenesis of autoimmune diseases through the engagement of additional genomic targets. ChIP-seq experiments with human B- and T-cells overexpressing BOB1 or having the BOB1/OCT1(2) axis disrupted with the help of our proprietary small molecule inhibitor (patent application WO2023180387A1) are currently underway.